# Are Personal Electric Vehicles Sustainable?
# A Hybrid E-Bike Case Study

**Mihai Machedon-Pisu ***  **and Paul Nicolae Borza**

Department of Electronics and Computers, Transilvania University of Brașov, 500036 Brașov, Romania;
borzapn@unitbv.ro
*** Correspondence: mihai_machedon@unitbv.ro; Tel.: +40-726-392-909

**Abstract:** As the title suggests, the sustainability of personal electric vehicles is in question. In terms of life span, range, comfort, and safety, electric vehicles, such as e-cars and e-buses, are much better than personal electric vehicles, such as e-bikes. However, electric vehicles present greater costs and increased energy consumption. Also, the impact on environment, health, and fitness is more negative than that of personal electric vehicles. Since transportation vehicles can benefit from hybrid electric storage solutions, we address the following question: Is it possible to reach a compromise between sustainability and technology constraints by implementing a low-cost hybrid personal electric vehicle with improved life span and range that is also green? Our methodology consists of life cycle assessment and performance analyses tackling the facets of the sustainability challenges (economy, society, and environment) and limitations of the electric storage solutions (dependent on technology and application) presented herein. The hybrid electric storage system of the proposed hybrid e-bike is made of batteries, supercapacitors, and corresponding power electronics, allowing the optimal control of power flows between the system's components and application's actuators. Our hybrid e-bike costs less than a normal e-bike (half or less), does not depend on battery operation for short periods of time (a few seconds), has better autonomy than most personal electric vehicles (more than 60 km), has a greater life span (a few years more than a normal e-bike), has better energy efficiency (more than 90%), and is much cleaner due to the reduced number of batteries replaced per life time (one instead of two or three).

**Keywords:** life cycle assessment; personal electric vehicles; hybrid energy storage system; e-bike

---

## 1. Introduction

In the context of the recent expansion of urban transportation systems, followed by rapid development of road infrastructure and motorization, the increase in the number of vehicles is completely justified [1,2]. Hopefully, in the near future, these will be seen as both smart and green transportation systems [3], and the corresponding infrastructure will be provided [4]. This could lead to the implementation of smart cities. Until that day, one has to wonder whether these transportation systems are sustainable [5–8]. In order to sustainably develop transportation systems, the following factors must be taken into account—economic growth, society demands, and environmental impact.

Figure 1 illustrates these sustainability challenges and their relation to transport systems, which need to be considered in order to develop sustainable, green and resilient cities [9]. Figure 1 also hints out at the need to use the appropriate methodology for implementing the vehicles of tomorrow (EVs), according to the sustainability challenges mentioned above. Going back in time, the first electric vehicles have appeared in the first half of the 19th century thanks to Aynos Jedlik in 1828 [10]. In 1834, Thomas Davenport built a small electric car [11]. Between 1832 and 1839, Robert Anderson invented the first crude electric vehicle powered by non-rechargeable chemical cells. In 1859, Gaston Planté

invented the first rechargeable battery (lead-acid), thus paving the way for electric cars. In 1901, Ferdinand Porsche invented the first hybrid electric car, the Lohner-Porsche Mixte. Since 1969, General Motors has been preoccupied on developing hybrid driving cars (see GM XP-883 in 1969), and in 1973, they proposed a hybrid car prototype due to pollution issues, based on the previous model of the Buick Skylark. This prototype was designed by Victor Wouk. In 1997, Toyota developed the Prius, the first mass production hybrid car.

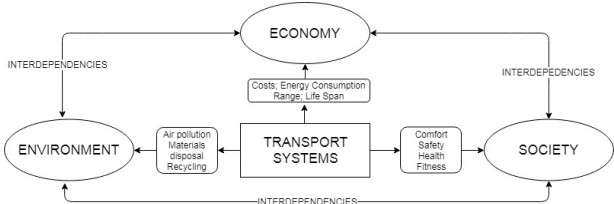

**Figure 1.** Sustainability challenges for transport systems.

The technologies for transport systems have always looked for compromises that can smooth the transition from motor vehicles with ICE to electric cars. Thus, "mild hybrid" cars have been developed, which are capable of reducing fuel consumption by 5–7%. This is done by recovering the braking energy and by optimizing the operating curves of the ICE as a result of adding a certain power supply from a small electric car [12]. The demand for energy storage on board has led to the increase in the standard voltage of the electrical energy sources energy per vehicle from 12 or 24 V to 48 V, respectively, above hundreds of volts for pure electric vehicles. Also, plug-in electric vehicles with increased range when operating in an electric regime have been proposed, being known as "extended ranger" systems. They can increase the range of the vehicle, but with a sacrifice in nominal performance [13].

The main types of EVs can be classified in two categories—(i) heavy-duty and medium EVs, like electric trams, trains, trucks, buses and e-cars, and (ii) light EVs (LEVs) or personal electric vehicles (PEVs), like electric motorcycles, scooters, and bikes. PEVs can be also split into medium battery EVs (BEVs), such as e-motorcycles, big e-scooters, and small BEVs, like e-bikes and small e-scooters.

Heavier EVs are made up of electric motors and batteries together with or without ICE or fuel cells, while smaller EVs, such as PEVs, use only motors and batteries and possibly also other storage elements. For example, e-bikes can be electric assisted, battery-based, or hybrid (e.g., batteries and supercapacitors (SCs), batteries and fuel cells (FCs)). A clear distinction should be made between these terms, as it is not always clear, as in reference [14]. Electric assisted bikes, or electrically power assisted bikes (EPAC), or pedelecs, get propulsion by means of an electric motor connected to batteries and still can be pedaled by the rider. E-motorcycles cannot be pedaled, as they are only battery-based. They are also known as BEVs. Hybrid e-bikes, or HESS bikes [15], which belong to the class of assisted hybrid vehicles (AHVs), use additional storage elements, like supercapacitors, to drive the motor when battery operation is not required.

Unfortunately, the current development of vehicles can have a damaging effect on the environment [16]. Air pollution caused by different classes of vehicles is discussed in references [17–20]. In order to reduce the effects of pollution and the dependence on oil, electric vehicles (EVs) should replace conventional cars (diesel and gasoline) in the future [21,22]. The high traffic can be diverted in large cities by opting for smaller EVs (PEVs and LEVs), such as electrical two-wheelers (e.g., e-bikes and e-scooters) that consume and pollute less, instead of heavier vehicles, such as cars and buses [23]. As discussed in reference [19], greenhouse gas (GHG) emissions are much lower for e-bikes than for other EVs. As seen in Table 1, e-bikes have less energy consumption, lower maintenance and purchase costs, require less travel time in congested traffic, and are beneficial to human health and fitness. E-bikes and other EVs present several limitations and issues such as lack of infrastructure and unitary regulations in safety, speed, and power; reduced autonomy and life span. But, crowded cities still require solutions that should imply green and sustainable smart transportation systems [16] that are

also beneficial to human health and fitness, or at least not damaging. Of course, the traditional bike is also green and does not consume any energy except for human effort. However, human effort is very limited, and older people, those with health issues, or those not fit enough can reduce the trip time significantly and cut the benefits of owning a bicycle. Some health benefits of e-bikes are presented in reference [24]. E-bikes and e-motorcycles are compared to the conventional ones in reference [25]. Reductions in human effort and variations of heart rate were reported for these small EVs. Reduction in travel time was also observed for e-bikes, but not for e-motorcycles. As mentioned in reference [26], e-bikes were reported with the shortest travel time in congested traffic: less than 5 km. It is also possible to transform a conventional bike into an electrical assisted bike, as mentioned in reference [27], but one has to think about costs and gain in performance, as you cannot always get the best of both worlds.

**Table 1.** Typical characteristics of different types of transportation vehicles.

| Characteristic | Cars, Trains, Trams, Trucks, Buses | Medium and Heavy-Duty EVs | Motorcycles, Big Scooters, Mopeds | Bikes, Small Scooters | E-Bikes |
|---|---|---|---|---|---|
| Air pollution | 4–30 [1] kilotons/year | 2–5 [1] kilotons/year | 10 [1] kilotons/year | Negligible * | 1–2 [1] kilotons/year |
| Health & fitness factor | No impact on fitness, no health benefits * | | | Beneficial to both [7] | |
| Infrastructure | Yes * | Conditional Yes: Road * | | Cond. Yes: Bike lane/road * | |
| Safety | High (industry regulations) to medium (when on road) * | | Medium (when on road) * | | Medium to low [3] (when on road + high speed) |
| Limits in speed, power, weight | No* | | mopeds: 45 km/h *, motorcycle: Tens of kW * | 100–300 W * | 25–32 km/h,100–750 W, <45kg [2] |
| Energy consumption | 10–40 MJ/user [4]; 2778–11,111 Wh/user | 5–15MJ/user [4]; 1389–4667Wh/user | 5–30MJ/user *; 1389–8333Wh/user | Negligible * | <1MJ/user *; 278Wh/user |
| Life span | 10–20 years * | 8–10 years* | | Up to 3–4 years * | |
| Life expectancy | >160,000 km *(cars); 0.5–1*10 [6] km (trucks) | >160,000 km *(e-cars); –(e-trucks) | 25,000–50,000 [8] km | 15,000 [8] km | 4000–15,000 [8] km |
| Recharging time | - | Tens of minutes to few hours ** | - | - | 5–8h[6] |
| Range | High: Up to thousands of km * (infrastructure & storage-dependent) | High to medium: A few hundreds of km *(storage-dependent, max. 500 km) | Medium: Daily use, hundreds of km * | - | Medium to low: Hours, 45–50 km [5] |
| Trip time | (Congested) 15–35 min [4]; (Light traffic) 5min * | | (Cong.) 15–20 min *; (Light tr.) 5–10 min * | 30min–1h * | 15–20 min [4] |
| Purchase costs | Tens/hundreds of thousands € * | | 3000 to 10,000 € * | below 1000€ * | 500 to 5000 € * |
| Maintenance costs | High: Up to tens of thousands € | | Medium: Hundreds to thousands of € * | Low to medium: Hundreds of € * | |

* According to estimations; ** dependent on charger and battery type; [1] according to references [2,17,19,20], based on average values for the sum of main pollutants: CO (carbon monoxide), NOx (nitrogen oxides), VOC (volatile organic compounds), and PM10 (particulate matter that have 10 micrograms per cubic meter or less in diameter); [2] according to reference [28]; [3] according to references [29,30]; [4] according to reference [26], based on average values for car/bus and e-bike, up to 5 km; [5] according to reference [21], based on average values; [6] according to references [31,32]; [7] according to reference [25]; [8] according to reference [33].

Performance is rather a complex task to quantify, as it comprises of various facets. Many studies have tried to cover a least a part of these facets, such as dynamic performance (speed, power, range and acceleration, according to local regulations, see Table 1) [17,21,31], energy consumption analysis (stored vs. used energy, autonomy, charging time, storage characteristics, see Tables 2 and 3) [21,25,26,33], functional performance (energy efficiency, temperature domain and other environmental parameters, route optimization, and minimizing daily accelerations and braking) [25,26], and sustainability (cyclability/durability that affects life span, state of charge characteristics) [29,34]. Most of these facets area direct result of the constraints on the design of EVs [35]. As seen in Figure 2, the e-Mobility design [36] is based on a reservoir of energy (energy source) powering an actual vehicle. Even if the design is simplified, its implementation is not that simple due the performance constraints discussed previously, which can become difficult to comply with.

**Table 2.** Life cycle assessment (LCA) for personal electric vehicles (PEVs) and medium battery electric vehicles (BEVs).

| Metric | Conventional Bike | E-Bike | BEV * | Conventional Motorcycle |
|---|---|---|---|---|
| Environmental impact (emissions and other products) | | | | |
| **Main Pollutants** | | | | |
| CO (g/km) | - | 0.02 [3] | 2.1–8.47 [2] | 1.7–5 [6]; 12.5–18 [3] |
| $NO_x$ (g/km) | - | 0.06 [3] | 0.11–0.37 [2] | 0.05–0.15 [3]; 0.1–0.34 [6] |
| HC (g/km) | - | - | 0.29–1.16 [2] | 0.56–4.69 [6] |
| PM10 (g/km) | - | 0.02 [3] | - | 0.06–0.33 [3] |
| PM2.5 (g/km) | - | 0.01 [3] | - | 0.03–0.16 [3] |
| $SO_2$ (g/km) | - | 0.13 [3] | - | 0 [3] |
| HC (g/km) | - | 0.007 [3] | - | - |
| $CO_2$ (g/km) | - | 21.5 [3] | 20.2–40.5 [3]; 30 [8] | 40–55 [3]; 68–105 [6] |
| **Production/Manufacturing Processes** | | | | |
| $SO_2$ (kg) | 0.275 [2] | 1.563 [2] | 2.198 [2] | 1 [2] |
| PM (kg) | 1.176 [2] | 5.824 [2] | 8.173 [2] | 4 [2] |
| GHG(tones SCE) | 0.097 [2] | 0.603 [2] | 0.875 [2] | 0.284 [2] |
| Waste water (kg) | 393 [2] | 1488 [2] | 2092 [2] | 1397 [2] |
| Solid waste (kg) | 0.641 [2] | 4.463 [2] | 7.139 [2] | 3 [2] |
| **Life Cycle of Lead Acid Battery Based Vehicles** | | | | |
| $CO_2$ (g/km) | 4.7 [3] | 15.6–31.2 [3] | 20.2–40.5 [3] | 64–128 [3] |
| $SO_2$ (g/km) | 0.01 [3] | 0.07–0.14 [3] | 0.09–0.17 [3] | 0.04–0.08 [3] |
| PM (g/km) | 0.06 [3] | 0.07–0.14 [3] | 0.1–0.19 [3] | 0.2–0.4 [3] |
| CO (g/km) | - | 0.007–0.014 [3] | 0.009–0.017 [3] | 6.3–12.5 [3] |
| HC (g km) | - | 0.027–0.053 [3] | 0.032–0.064 [3] | 1.13–2.25 [3] |
| $NO_x$ (g/km) | - | 0.01–0.02 [3] | 0.014–0.027 [3] | 0.08–0.15 [3] |
| Pb (mg/km) | 0 [3] | 145–290 [3] | 210–420 [3] | 16–32 [3] |
| Energy impact | | | | |
| **Energy Used When Operating** | | | | |
| Energy consumption (MJ/user) | 0.25 [10]; 69.44 Wh | 0.19 [9]–0.52 [1]; 52.77–144.4 Wh | 4.68–14.97 [1]; 1300–4160 Wh | - |
| Energy per km (MJ/km) | 0.013 [1]; 3.62 Wh/km | 0.028 [1]; 7.78 Wh/km | 0.73 [9]; 203 Wh/km | 0.67–0.85 [9]; 186–236 Wh/km |
| Energy use (kWh/100 pax-km) | 4.88 [3] | 3.8–7.6 [3] | 4.9–9.9 [3]; 5.7 [7] | 21–42 [3] |
| Battery (kWh) | - | 0.36 [9] | 1.68–5.4 [9] | - |
| **Energy Used Per Activity(KJ/PKT)** | | | | |
| Fuel production | 0 [5] | 1.25 [5] | - | 50–150 ** |
| Infrastructure | 126 [5] | 126 [5] | - | 200–500 ** |
| Maintenance | 5.5 [5] | 5.5 [5] | - | 60–150 ** |
| Manufacturing | 66 [5] | 87 [5] | - | 140–200 ** |
| Operation | 0 [5] | 0 [5] | - | 600–700 ** |
| **Energy Used Per Life Cycle (MJ)** | | | | |
| Manufacture | - | 12,000 [4] | 20,000 [4] | - |
| Use | - | 87,000 [4] | 265,000 [4] | - |
| Disposal | - | 1200 [4] | 3000 [4] | - |
| Life cycle energy consumption | - | 102,000 [4] | 288,000 [4] | - |

* E-cars and small e-scooters are excluded, ** according to estimations, [1] according to Reference [21] for Well-to-Wheel (WTW) analysis, [2] according to Reference [17] for test scenarios in Italy, [3] according to Reference [23] for the models used, lead losses are considered for a recycle rate from 100 to 90%, [4] according to Reference [37], [5] according to Reference [38], PKT is passenger occupancy-km-traveled, [6] according to Reference [18] for test scenarios in Switzerland, [7] according to Reference [39], [8] according to Reference [40], [9] according to Reference [25], [10] according to Reference [41].

**Table 3.** Environmental impact of lead-acid battery-based vehicles—materials and battery losses.

| Metric | Conventional Bike | E-Bike | BEV * | Conventional Motorcycle |
|---|---|---|---|---|
| **Weight of materials used** | | | | |
| Total weight (kg) | 15 [6];17 [3]; 18 [2] | 23 [4]–24 [3]; 26 [1]–41 [2] | 65.8 [2]; 140 [4]–144 [3]; 80–208 [5] | 90 [3,4]; 94 [2] |
| Steel | 13 [2] | 18.2 [2] | 26.2[2] | 76.4 [2] |
| Plastic | 2 [2] | 5.7 [2] | 15.2[2] | 9.1 [2] |
| Lead | - | 10.3 [2] | 14.7[2] | 1.7 [2] |
| Nickel | - | - | - | 0.3 [3] |
| Fluid | - | 2.9[2] | 4.2[2] | - |
| Copper | - | 2.6[2] | 3.5[2] | 1 [2] |
| Rubber | 2 [2] | 1.1 [2] | 1.2 [2] | 3.2 [2] |
| Aluminum | 1 [2] | 0.5 [2] | 0.6 [2] | 1.5 [2] |
| Maintenance | 50%Plastic, 5%Steel [4] | 50%Plastic, 5%Steel [4] | 10%Steel, 10%Aluminum [4] | 10%Steel, 10%Aluminum [4] |
| **Lead Acid Battery Losses (kg Per Battery)** | | | | |
| Battery weight | - | 10.3 kg [2] | 14.7 kg [2]; 32 kg [4] | 1.7 kg [2] |
| Mining and concentration loss | - | 1.1–1.2 [2] | 1.5–1.7[2] | 0.17–0.19 [2] |
| Smelt loss (primary) | - | 0.4 [2] | 0.6 [2] | 0.06–0.07 [2] |
| Smelt loss (sec.) | - | 0.9–1 [2] | 1.3–1.4 [2] | 0.14–0.16 [2] |
| Manufacture loss | - | 0.5 [2] | 0.7 [2] | 0.08[2] |
| Total production emissions | - | 2.9–3 [2] | 4.2–4.3 [2] | 0.48–0.49 [2] |
| Solid waste | - | 0–1 [2] | 0–1.5 [2] | 0–0.17 [2] |

* E-cars and small e-scooters are excluded, [1] according to reference [17] for test scenarios in Italy, [2] according to reference [23] for the models used, lead losses are considered for a recycle rate from 100 to 90%, [3] according to reference [33], [4] according to reference [39], [5] according to reference [25], [6] according to reference [41].

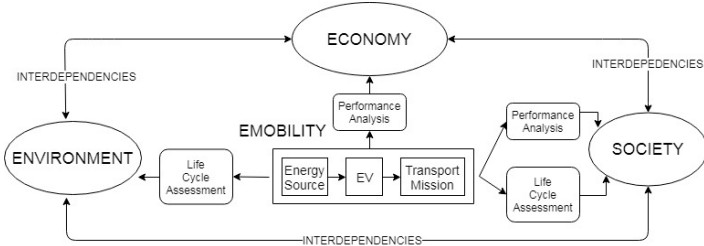

**Figure 2.** Methodology for developing sustainable EVs (e-Mobility).

Starting from Figure 2, state the research question of this paper is whether it possible to provide an appropriate methodology for implementing a sustainable design of hybrid electric transportation systems in the context of current storage solutions (Li-ion batteries, SC, FC) [42] in order to increase their autonomy, on one hand, and reduce the costs and impact on environment, on the other. In order to provide an answer, the paper is organized as in Figure 3.

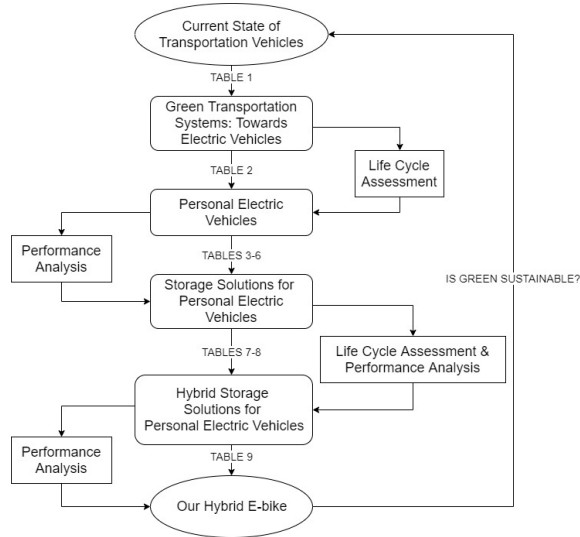

**Figure 3.** Methodology proposed in paper for developing sustainable PEVs.

In the introduction, we presented the current state of vehicles in terms of sustainability, emphasizing on the vehicles of the future—EVs, as justified in Table 1. In Section 2, the performance of small and medium EVs, which include PEVs, is analyzed in terms of life cycle assessment (LCA), thus addressing the society requirements (and the need for compromise) for building green transportation systems, with less lead batteries, as justified in Tables 2 and 3. Section 3 presents the characteristics of the storage elements used for such EVs underlining their limitations, presented in Tables 4–6. Section 4 analyzes the possibility to employ a hybrid storage system based on supercapacitors (SC) and/or fuel cells for PEVs in terms of LCA, as seen in Table 7. This section and the next emphasize the methods that can provide the right balance between the benefits and limitations of such storage elements (SC and FC) in conjunction with Li-Ion batteries, in order to implement hybrid storage systems for PEVs, as illustrated in Table 8. Section 5 proposes a low-cost hybrid bike that tackles most sustainability issues of PEVs, as demonstrated in Table 9. Section 6 presents and discusses the guidelines for designing green and sustainable vehicles containing hybrid electric storage systems (HESS), based on the findings of the paper. Section 7 presents the main findings of this paper, related to the used methodology for designing sustainable PEVs and the performance of our solution for PEVs—a hybrid e-bike with SC and Li-Ion battery storage.

## 2. Life Cycle Assessment

Apart from the immediate consequences of safety, health, range, and cost on EV usage, aspects already presented in Table 1, another aspect, life cycle assessment (LCA) is relevant for EVs, especially in terms of environmental impact and energy consumption. PEVs consume less energy and cause less pollution than heavier EVs.

Table 2 presents the LCA impact of PEVs and can be seen as an extension of Table 1. Although this type of analysis implies some uncertainty due to changes in technology, rider behavior [38], lack of standard methodology, and metrics for dynamic performance [33,39], it can still be used for testing the sustainability of an EV, but only if it is correlated with other types of analysis that can guarantee the static and dynamic performance metrics related to the application's requirements. Today, from the application point of view, the advances in power electronics permit cheap and very efficient commutation solutions [36]. In this sense, PEVs could be seen as the right candidate for testing the control strategies of EVs at a smaller scale. In order to meet the sustainability challenges, e-bikes, which are one of the most promising PEVs, permit the appropriate compromise between most performance metrics due to their customizable design.

A European study analyzed the economic and social impact of owning an e-bike, in reference [43], reporting fuel savings of ~300 € per year, an acceptance rate of 70%, and a safe operation since no incidents were reported.

## 3. Performance Analysis of Main Storage Solutions for PEVs

As can be observed in Figure 4, when comparing lead-acid battery–based conventional motorcycles to e-bikes and heavier PEVs, such as e-scooters, the latter have a more negative impact on life cycle in terms of Pb (mg/km) and $SO_2$ (mg/km) [23]. E-bikes and other PEVs have a short life span, and therefore the disposal of batteries can have a very negative impact on environment [44]. When it comes to energy consumption, energy use (kWh) has the largest impact, as seen in Figure 5. This can be attributed to the influence of EV storage. Table 4 analyzes the main performance metrics of the most used storage elements that provide the necessary energy and power for the propulsion of PEVs.

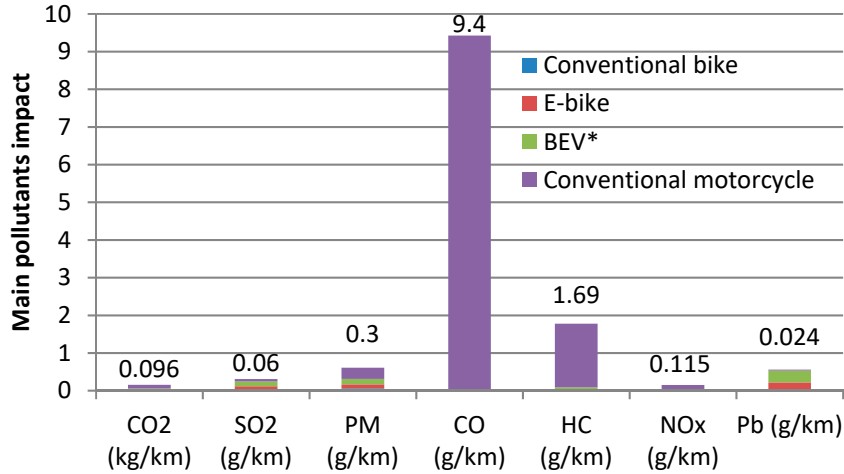

**Figure 4.** Environmental impact of lead-acid battery–based transportation vehicles, according to the average values from Table 2. The values on top are for a conventional motorcycle. * E-cars and small e-scooters are excluded.

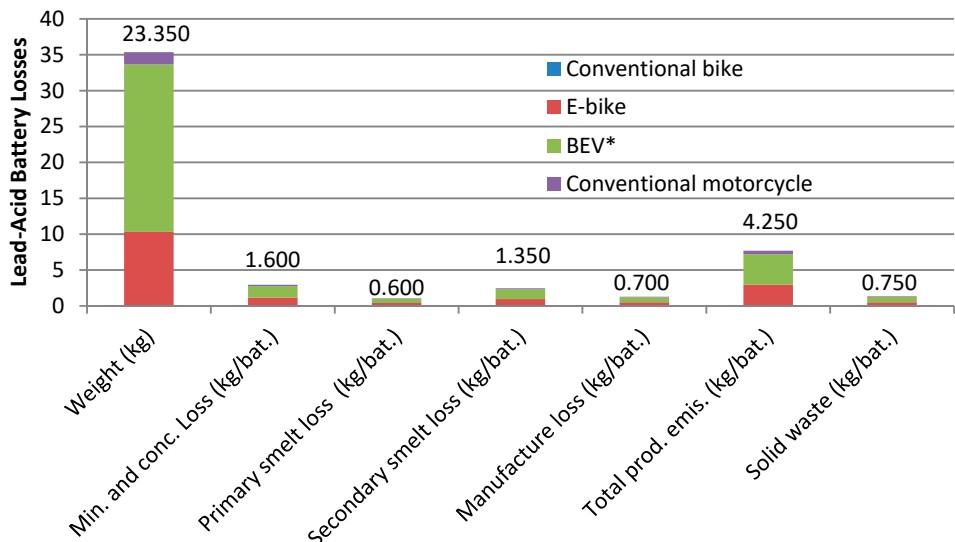

**Figure 5.** Battery losses (kg per battery) of lead-acid battery–based transportation vehicles, according to the average values from Table 3. The values on top are for a BEV. * E-cars and small e-scooters are excluded.

One storage solution that is absent from Table 4 is Li polymer. It is a promising lithium battery that presents good power density, but their calendar life is modest, as detailed in reference [45]. There are also other types of battery and storage solutions that were not considered in Table 4. Ni-iron, Ni-zinc, Ni-cadmium, aluminum-air, and zinc-air batteries are analyzed in reference [46]. Nickel's low operating voltage is similar with that of Ni-MH, while Zinc-air has a reduced number of life cycles. Li-oxygen, Li-sulfur [46], and magnesium-ion batteries are even better than Li-ion batteries in energy density, as discussed in [47], in the 300–1000 Wh/kg range. However, these are only predicted values since they are not commercially available. Other types of storage—pumped hydroelectric storage (PHS), compressed air energy storage (CAES), flywheels, capacitors, sodium-sulfur (NaS) batteries, vanadium redox (VRB), zinc-bromine (ZnBr) and polysulfide bromine (PSB) flow batteries, superconducting magnetic energy storage (SMES), solar fuel, thermal energy storage (TES), and liquid air storage are analyzed in reference [34]. Most of these solutions are incompatible to EVs both in terms of costs and deployment issues, as well as (lack of) maturity, or they have poor performance, such as energy or power density.

**Table 4.** Characteristics of the main storage solutions for PEVs.

| Characteristic | Lead Acid | Li Ion | Supercapacitors | Hydrogen Fuel Cells | Ni-MH & Ni-Cd |
|---|---|---|---|---|---|
| Energy density (Wh/l) | 50–70 [1]; 50–90 [4] | 150–200 [1]; 150–500 [4] | 10–30 [4]; 3–180 [6] | 500–3000 [4] | 200 [1]; 170–420 [4] |
| Power density (W/l) | 10–400 [4] | 1500–10,000 [4] | 100,000+ [4] | 500+ [4] | 80–600 [4] |
| Specific energy (Wh/kg) | 20–40 [1]; 35 [2]; 25–50 [4] | 100–200 [1];<120–150 [2]; >200 [3]; 75–200 [4]; 150–350 [5] | 0.05–15 [4] | 100 [5]–10,000 [4] | 40–60 [1]; <70 [2]; 70–100 [4]; 15–300 [5] |
| Specific power (W/kg) | 300 [1]; 150–900 [2]; 75–300 [4,5] | 300–800 [1]; <120–150 [2]; <150–2000 [4] | 500–10,000 [4] | 5–800 [4]; 500 [5] | 130–500 [1]; <200 [2] |
| Power range (MW) | 0–40 [4]; <20 [5] | 0–100 [4]; <0.001 [5] | 0–0.3 [4] | 10–58.8 [4]; 0.3–50 [5] | 0–40 [4,5] |
| Rated energy capacity (MW h) | 0.001–40 [4] | 0.004–10 [4] | 0.0005 [4] | 0.3–39 [4] | 6.75 [4] |
| Voltage (V) | 2.1 [1,2] | 3.6 [1,2] | 2.3–2.8 [6] | - | 1.2 [1,2] |
| Overall efficiency (%) | 85 [1]; 70–90 [5] | 93 [1]; 85–95 [5] | 82–98 [5] | 33–42 [5] | 60–73 [5]; 80 [1] |
| Cycle efficiency (%) | 63–90 [4] | 75–97 [4] | 84–97 [4] | 20–66 [4] | 60–83 [4] |
| Discharge efficiency (%) | 85 [4] | 85 [4] | 95–98+ [4] | 59 [4] | 85 [4] |
| Cycle life @80%DOD | 200 [1]; 500–1000 [2] | < 2500 [1]; >1000 [2] | 100,000–1,000,000 [6] | - | > 2500 [1]; >2000 [2] |
| Lifetime (years) | 5–15 [4,5] | 5 [5]–16 [4] | 10–30 [4] | 5–20+ [4] | 3 [4]–20 [5] |
| Life cycles (cycles) | 200–1800 [4]; 2000–4500 [5] | 1000–20,000 [4]; 1500–4500 [5] | >50,000 [5]; >100,000 [4] | 1000–20,000+ [4] | 2000 [5]–3500 [4] |
| Self-discharge (%/Day) | 0.1–0.3 [4,5] | 0.1 [5]–5 [4] | 5–40 [4] | Almost 0 [4,5] | 0.03–0.6 [4] |
| Fastest 80% recharge time (min) | 15 [2] | <60 [2] | - | - | 35 [2] |
| Response time | | Milliseconds, 1/4 cycle [4] | | Seconds, <1/4 cycle [4,*] | Milliseconds, <1/4 cycle [4] |
| Suitable storage duration | Minutes-days (short to medium term) [4,5] | Minutes-days (short to med. term) [4,5] | Seconds-hours (short term, <1 h) [4] | Hours-months [4,**] | Minutes-days [4,5] |
| Discharge time at power rating | seconds-hours (up to 10 h) [4,5] | minutes-hours (1–8 h) [4,5] | miliseconds-1 h [4] | seconds-24 h+ [4] | seconds-hours (1–8 h) [4,5] |
| Operating and maintenance cost | 50 $/kW/year [4] | - | 0.005–6 $/kW-year [4] | 0.0019–0.0153 $/kW-year [4] | 50 $/kW/year [4] |
| Total capital cost, per unit of power rating (€/kW) | 1388–3254 [5] | 2109–2746 [5] | 214–247 [5] | 2395–4674 [5] | 2279–4182 [5] |
| Total capital cost, per unit of storage capacity (€/kWh) | 346–721 [5] | 456–560 [5] | 691–856 [5] | 399–779 [5] | 596–808 [5] |
| Maturity | Mature [4] | Demonstrated [4] | Demo./Developing [4] | Demo./Developing [4] | Demo. [4] |

* Response time refers to reactivity of fuel cells (FCs) after the startup period, which depends on the FC type, ** storage duration depends on the reservoir capacity, [1] according to reference [48], [2] according to reference [49], [3] according to reference [47], [4] according to reference [34], [5] according to reference [50], [6] according to reference [51].

**Table 5.** Typical characteristics of other lithium and nickel based batteries.

| Characteristic | NCA — Nickel Cobalt Aluminum Oxide | LMO/LTO — Lithium Manganese/Titanium Oxide | LFP — Lithium Iron Phosphate | NMC — Nickel Manganese Cobalt Oxide | LMO-NMC — - |
|---|---|---|---|---|---|
| Specific energy (Wh/kg) | 200–260 [2] | 50–80 [2] | 90–120 [2] | 150–220 [2] | - |
| Energy density (Wh/kg) | 130 [7] | 85 [7] (LTO), 114 [3]–120 [7] (LMO) | 93 [3]–130 [7] (poor [8]) | 170 [7] | 120–170 * |
| Power | Acceptable [7]: 100–200 W | Good [7]: 200–500 W (LTO), Acceptable [7] | Acceptable [7] | Average [7]: 50–100 W, AccepTable [8] | - |
| Energy consumption (Wh/km) | - | 105–214 [3] | 114–223 [3] | - | - |
| Energy capacity (kWh) | – | 24–34.2 [1] | 18.5 [4,5]–24 [3] | 26.6 [1];63.5 [3] | 24 [1] |
| Nominal capacity (Ah) | - | - | 60 [4]; 40 [9]; 2.3 [10]; 90 [12] | 2.3–12.4 [5];40–50 [6] | - |
| Nominal current (A) | - | - | 18.3 [11]; 40 [4] | - | - |
| Nominal voltage (V) | 3.6 [2]; 3.65 [7] | 2.4 [2,7] (LTO); 3.8 [2]–4 [7] (LMO) | 3.2 [2,3]; 3.3 [2] | 3.6 (3.7) [2]; 3.8–4 [7] | 3.6–4 * |
| Charge (C-rate) | 0.7 C, (4.2 V), typical charge time 3 h [2] | 1 C, (2.85 V) [2] | 1 C typical, max.10 C [13] (3.65 V [4]), typical charge time 3 h [2] | 0.7–1 C, (4.2 to 4.3 V), typical charge time 3 h [2] | - |
| Discharge (C-rate) | 1 C (3 V) [2] | 10 C (1.8 V) [2] | 1 C (2.5 V) [2,4]; max.5 C-15 C [14] | 1 C, 2 C (2.5 V) [2] | - |
| Battery efficiency (%) | - | 95 [1] | 82.3 (1.2 C)–94.5 (0.1 C) [4] | 95–96 [1] | 95–96 * |
| Depth of discharge (%) | - | 70 [3] | 70 [3] | - | - |
| Cycle life | 500 [2] | 2000–25,000 [2]; 1400–1500 [3] | 1000–2000 [2] | 1000–2000 [2]; 1500 [5] | - |
| Cost ($ per kWh) | ~350 [2] | ~1005 [2] | ~580 [2] | ~420 [2] | - |
| Safety | Average [7], poor [8] | Good [7] (LTO), acceptable [7] | Good [7], acceptable [8] | Average [7], poor [8] | - |

* According to estimations, [1] according to reference [52], [2] according to reference [53], [3] according to reference [54], [4] according to reference [55], [5] according to reference [56], [6] according to reference [57], [7] according to reference [49], [8] according to reference [45], [9] according to reference [58], [10] according to reference [59], [11] according to reference [60], [12] according to reference [61], [13] according to reference [62], [14] according to references [62–64].

**Table 6.** Typical characteristics of supercapacitors, compared to Li-ion batteries and fuel cells.

| Characteristic | Supercapacitors (SC) | | | Li-Ion Battery | Hydrogen Fuel Cells |
| | EDLC SC | Pseudo SC | Hybrid SCAsymmetric | | |
| --- | --- | --- | --- | --- | --- |
| Type of electrolyte | Aprotic or protic [1] | Protic [1] | Aprotic [1] | Aprotic [1] | - |
| Energy density (Wh/kg) | 5–20 [4]; 3–5 [1]; <6.5 [2] | 10 [1]; <25 [2] | 180 [1]; 20–30 [2]; <125 [2] | 250 [1]; 120–200 [2] | 100 [3]–10,000 [6] |
| Power density (W/kg) | 1500 [4]; Up to 6000 [2] | Up to 6000 [2] | 10–1000 [2] | 300–800 [4]; <120–150 [5]; <150–2000 [6] | 5–800 [6]; 500 [3] |
| Cell voltage (V) | 2.5 [4]; 2.7 [1] | 2.3–2.8 [1] | 2.3–2.8 [1] | 3.6 [1] | - |
| Charge time (s) | 1–10 [1] | 1–10 [1] | 100 [1] | 600 [1] | - |
| Life Cycles | 1,000,000 [1] | 100,000 [1] | 500,000 [1] | 500 [1]; <2500 [4]; >1000 [5] | 1000–20,000+ [6] |
| Overall efficiency (%) | 97 [4]; 82–98 [3] | 82–98 [3] | <90 [2] | 85–95 [4] | 33–42 [3] |
| Self discharge per month (%) | 30 [4]; 60 [1] | 60 [1] | - | 4 [1]; 1–5 [4] | 20 [4] |
| Temperature of operation (°C) | −30–65 [4]; −40–65 [1] | −40–65 [1] | −40–65 [1] | −20–60 [1]; −20–55 [4] | - |
| Cost per kWh ($) | ~10,000 [1]; <1000 [3]; 2200 [4] | ~10,000 [1]; <1000 [3] | <1000 [3]; 300–2000 [3] | 140 [1]; 500–600 [3]; 800 [4]; 150 [5]; 600–3800 [6] | 450–900 [3] |
| Cost per kW ($) | 55 [4]; 100–450 [6] | 55 [5–4]; 100–450 [6] | 55 [4]; 100–450 [6] | 55–80 [5]; 900–4000 [6] | 20 [4]; 500–1500 [6] |

[1] According to Reference [51], [2] according to reference [65], [3] according to reference [50], [4] according to reference [48], [5] according to reference [49], [6] according to reference [34].

Figure 6 provides a visual representation of the power and energy densities of the main types of storage, according to the characteristics detailed in Table 4, emphasizing on range (cycle life) associated to specific energy, and acceleration, associated to specific power.

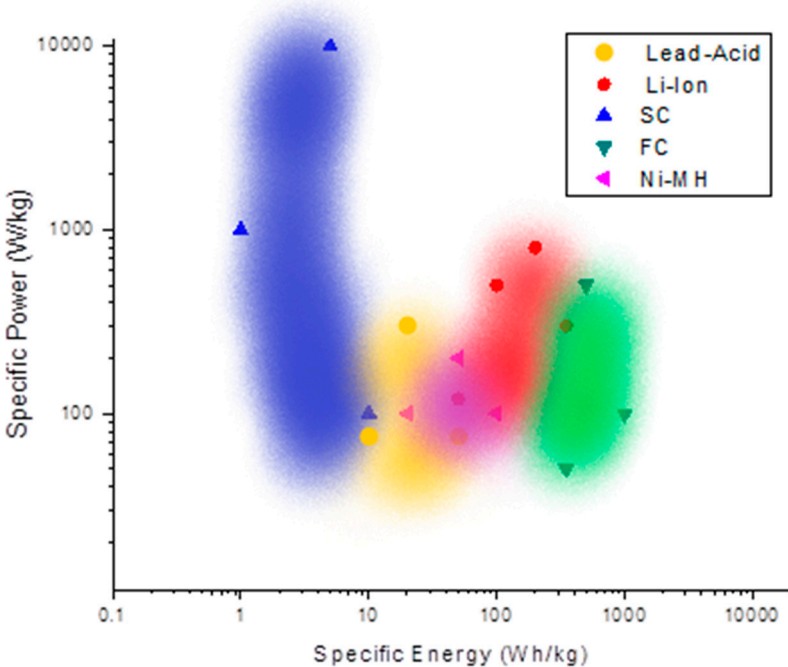

**Figure 6.** Relation between specific power and energy (Ragone plot) for the main types of storage for PEVs, according to the average values from Table 4.

Table 4 presents both the disadvantages and advantages of the batteries used for providing propulsion to small EVs. In Table 5, the performance of other types of lithium and nickel batteries is analyzed.

Due to the increased weight of lead-acid-based batteries, the losses associated to lead can become 5 to 10 times greater for smaller EVs than for conventional motorcycles, as seen in Table 3. In a similar fashion, the losses associated to lithium-based and nickel-based batteries, presented in Tables 4 and 5, can become very relevant in terms of environmental impact. As recommended in [29], for a large-scale deployment of EVs, less lithium must be used per unit of battery storage, or a suitable type of energy storage system that does not use lithium must be developed. This is justified by the cumulative demands of EV/PHEVs that could exhaust the whole lithium reserve by 2050, even with extensive recycling. The main alternatives to batteries are supercapacitors (SC) and hydrogen fuel cells (FC). In Table 6, their characteristics are compared to Li-ion batteries.

Due to the low operating voltage, symmetric hybrid supercapacitors were not considered for analysis in Table 6, as discussed in reference [65]. Also, according to reference [66], Li-ion batteries cannot withstand more than 1000 cycles per lifetime, which contradicts many other studies that claim thousands of life cycles.

**Table 7.** LCA of Li-ion battery only, FC only and hybrid (battery-FC) bikes.

| Metric | | Li-Ion Battery | Fuel Cells |
|---|---|---|---|
| **Health (1) and Environmental Impact (2) for Production Phase** | | | |
| | Unit | E-bike | Hydrogen bike |
| (1) Carcinogens | kg C2H3Cleq | 0.0028 [1] | 0.003 [1] |
| (1) Non-carcinogens | kg C2H3Cleq | 0.0035 [1] | 0.0035 [1] |
| (1) Respiratory inorganics | kg PM2.5eq | 0.039 [1] | 0.051 [1] |
| (1) Ionizing radiation | Bq C-14 eq | Below 0.0001 [1] | Below 0.0001 [1] |
| (1) Ozone layer depletion | kg CFC-11 eq | Below 0.0001 [1] | Below 0.0001 [1] |
| (1) Respiratory organics | kg C2H4eq | Below 0.0001 [1] | Below 0.0001 [1] |
| (1) Human toxicity | kg 1.4-DB eq | 230 [1] | 581 [1] |
| (1) Particulate matter formation | kg PM10 eq | 0.52 [1] | 1.04 [1] |
| (1) Photochemical oxidant formation | kg NMVOC | 0.67 [1] | 1.21 [1] |
| (2) Aquatic ecotoxicity | kg TEG water | Below 0.0001 [1] | Below 0.0001 [1] |
| (2) Terrestrial ecotoxicity | kg TEG soil | 0.0035 [1] | 0.0039 [1] |
| (2) Terrestrial acid/nutri | kg SO2 eq | 0.0005 [1] | 0.0007 [1] |
| (2) Land occupation | m2org.arable | 0.0002 [1] | 0.0002 [1] |
| (2) Aquatic acidification | kg SO2 eq | 0.0001 [1] | 0.0001 [1] |
| (2) Aquatic eutrophication | kg PO4 P-lim | 0.0001 [1] | 0.0001 [1] |
| (2) Fossil depletion | kg oil eq | 41.2 [1] | 66.2 [1] |
| (2) Metal depletion | kg Fe eq | 118.5 [1] | 176 [1] |
| **Climate Change (3) and Resources (4) for Production Phase** | | | |
| (3) Global warming | kg $CO_2$ eq | 0.02 [1]; 165.2 [2] | 0.023 [1]; 276.35 [2] |
| (4) Non-renewable energy | MJ primary | 0.0155 [1] | 0.018 [1] |
| (4) Mineral extraction | MJ surplus | 0.0005 [1] | 0.0013 [1] |
| **Health (1), Environmental Impact (2), and Climate Change (3) for Use Phase** | | | |
| | Hybrid e-bike (Battery-FC) | E-bike | Hydrogen bike (FC) |
| (1) Photochemical oxidant formation | 0.002 [2] | 0.004 [2] | 0.001 [2] |
| (1) Particulate matter formation | 0.002 [2] | 0.003 [2] | 0.001 [2] |
| (1) Human toxicity | 0.9 [2] | 1.07 [2] | 0.55 [2] |
| (2) Fossil depletion | 0.2 [2] | 0.4 [2] | 0.07 [2] |
| (2) Metal depletion | 0.49 [2] | 0.51 [2] | 0.31 [2] |
| (3) Global warming | 0.8 [2] | 1.42 [2] | 0.31 [2] |

[1] According to Reference [35], [2] according to Reference [67].

## 4. Life Cycle Assessment of Hybrid Storage Implementations/Solutions

One approach to avoid battery losses is to use other storage elements, such as FC and SC. For example, by using hydrogen fuel cells, one can implement a hydrogen bike. An e-bike is compared with a hydrogen bike in terms of environment, health and energy impact in Table 7.

The possibility to develop a hybrid e-bike, based on fuel cells and batteries, is analyzed in Table 7. In reference [68], different stages of hybridization for a 54 kW light electric vehicle (LEV) are discussed in terms of costs. The storage solutions and combinations include three storage elements: batteries, supercapacitors and hydrogen fuel cells. The costs range from approximately $23,000–34,000 USD, depending on the hybrid combination (SC-FC, battery-FC, and SC-battery) or standalone storage solution (FC and battery). The lowest prices were obtained for the hybrid storage implementations (battery-FC and SC-FC) and the highest for the FC implementation.

**Table 8.** Performance analysis of hybrid bikes compared to other PEVs and BEVs.

| Metric | Hybrid E-Bikes (SC-Battery and FC-Battery) | E-Bike (Pedelec/Battery only, SC only) and Small E-Scooter | Big E-Scooter and E-Motorcycle (Battery only) |
|---|---|---|---|
| Specific energy (Wh/kg) | - | 32.7–51.4 [1] | - |
| Energy expenditure (Wh/km) | - | 59.8 [2] (CB:69.8 [2]); 6.92–8.57 [6] | 202.86 [2] (big EM) |
| Battery energy (MJ) | 4.52 [19] (SC-bat) | 0.25–0.52 [12] (CB: 0.12 [12]); 4.67 [18] (SC only) | 5 [12] (LEV) |
| SC energy (MJ) | 0.071 [13] (SC only) | | - |
| Overall energy (Wh) | - | 155 [1]–360 [1,2]; 160 [8] | 1680–2880 [2]; 5400 [2] (big EM) |
| Power (W) | max.693 [20] (SC-bat) | 250 [7] (small e-scooter); 250 [1,3,8,9]–800 [1]; max.: 539 [3]; max.: 731–950 [11]; 143–1018 [15]; 2 000 [14] (big e-bike); 150–500 [13] (SC only); | 2000 [2]–6000 [2,21]; 20,020 [2] (big EM) |
| Battery capacity (Ah) | 12 [19] (SC-bat) | 5.2 [7] (small e-scooter); 5.4 [1]–10 [1,3,9]; 75 [20] | 40–80 [21] (big e-scooter); |
| Voltage (V) | 15–48 [18]; 12–16 [20] (SC-bat) | 48 [3]; 30 [8]; 36 [1,9,10]; 29.6 [1]; 12.73 [20]; 70–78 [13] (SC only); | 74 [21] (big e-scooter) |
| Current (A) | 8.8 [16] (FC-Bat); 12.23 [19] (SC-bat) | 18.39 [19] | - |
| Charging time (hours) | - | 5 [3] | - |
| Life Cycles | SC: 100,000–1,000,000 [7]; Li-Ion Battery: 500 [4]–800 [3] | Li-Ion Battery:500 [4]–800 [3] | - |
| Life expectancy (km) | 15,000 [4] | 15,000 [4,5]; 24,000 [22] | 50,000 [4,5] |
| Trip autonomy (km) | - | 37–55 [2]; 46–82 [12] (CB: real, 4–8 [2]); real:25–30 [10]; 13–80 [13] (SC only) | 26.6 [2] (big EM); 100 [21] (big e-scooter) |
| Battery type | Li-Ion [16] (FC-Bat) | Li-Po [3]; Li-Ion [13] | Lithium-based, Lead-based [2] |
| Total weight (kg) | 23 [5]; 27.1 [16] (FC-Bat) | 18 [19]; 23 [2]–26 [3,2]; 20.2–28 [1]; 41.3–65.8 [17] (big e-bike) | 90–144 [3]; ~140 [5]; 208 [2] (big EM) |
| Weight ratio (vehicle/80 kg rider) | 0.28–0.34 * | 0.23–0.35 *; 0.51–0.82 * (big e-bike) | 1.12–1.8 *; 2.6 * (big EM) |

* According to estimations and average values, [1] according to reference [69], [2] according to reference [25], [3] according to reference [17], for the tested pedelec, [4] according to reference [33], [5] according to reference [39], for the tested models, [6] according to reference [70], [7] according to reference [49], [8] according to reference [71], [9] according to reference [22], [10] according to reference [72], [11] according to reference [73], [12] according to reference [21], [13] according to reference [74], [14] according to reference [75], [15] according to reference [76], [16] according to reference [77], [17] according to reference [23], [18] according to reference [58], [19] according to reference [60], [20] according to reference [78], [21] according to reference [79], [22] according to reference [67].

**Table 9.** Performance analysis and costs of our hybrid bike (SC-battery) compared to other implementations.

| Metric | Our Hybrid E-Bike | E-Bike (in %) | Big E-Scooter and E-Motorcycle (in %) |
|---|---|---|---|
| Battery energy (MJ) | 2.190; 608.33 Wh | 12–24% [1](CB: 6% [1]) | 228% [1] (LEV) |
| SC energy (MJ) | 0.0148; 4.11 Wh | - | - |
| Overall energy (Wh) | 612.44 | 26 [10]–59% [2,10] | 276–473% [2]; 887% [2] (big EM) |
| Maximum power (W) | 1800 | 8–56.5% [12]; 14% [6](small e-scooter); 111% [11] (big e-bike) | 111 [2]–333% [2,15]; 1112% [2] (big e-motorcycle) |
| Battery capacity (Ah) | 13 | 42 [10]–77% [3,8,10]; 40% [6] (small e-scooter) | 308–616% [15] (big e-scooter) |
| Voltage (V) | 46.8 | 64 [10]–102% [3] | 158% [15] (big e-scooter) |
| Current (A) | 42 * | 44% [14] | - |
| Trip autonomy (km) | ~70* | 53–78.5% [2]; 66–102% [1] (CB: real, 10% [2]); real:40% [9] | 38% [2] (big EM); 142% [15] (big e-scooter) |
| Battery type | Li-Ion | Li-Po [3]; Li-Ion [10] | Lithium-based, Lead-based [2] |
| Number of batteries replaced per lifetime | 1 | 275% [5] | 100% [4] |
| Life time (years) | 2–4 ** | 33–50% ** | - |
| Bike incl. chassis ($& kg) | 200$; 11.6 kg | 144% in kg [5] | 861% in kg [5] |
| Batteries ($& kg) | 250$; 3.1 kg | 84 [5]–97 [9] % in kg; 332 [13] % in kg | 1030% in kg [5] |
| SC + DC/DC converter ($& kg) | 250$;1.3 kg | - | - |
| Motor + AC/DC converter($& kg) | 400$; 11.5 kg | 18–22% in kg [5] | 46–56% in kg [5] |
| Total cost ($) | 1100 | ~200% [7] | - |
| Total weight (kg) | 27.5 | 65–101% [10]; 140–234% [13] (big e-bike) | 324–518% [3]; 720% [2] (big EM) |

* Limited by power electronics, ** according to estimations and average values, [1] according to reference [21], [2] according to reference [25], [3] according to reference [17], for the tested pedelec, [4] according to reference [33], [5] according to reference [39], for the tested models, [6] according to reference [80], [7] according to reference [27], [8] according to reference [18], [9] according to reference [72], [10] according to reference [69], [11] according to reference [75], [12] according to reference [76], [13] according to reference [23], for a lead-acid e-bike, [14] according to reference [60], [15] according to reference [79].

## 5. Hybrid E-Bike Sizing and Performance Analysis

As discussed in reference [68] and shown in the last rows of Table 7, EV operation can be split between the battery and other storage elements. This is the second approach that aims to reduce the battery dependence. Table 8 compares hybrid e-bikes to e-bikes, e-scooters, and e-motorcycles in terms of performance. According to the eco-indicators detailed in [39], both e-bikes and conventional bikes have the lowest impact on environment. One of the main reasons for these results can be attributed to the reduced demand of energy of these vehicles. E-bikes also have a battery disposal impact that is two to three times higher than that of big and medium e-scooters. In order to reduce the battery dependency even more, hybrid e-bikes could be seen as a better alternative.

Most e-bikes (pedelecs) present the following features: the motor is placed on the rear wheel [17], have a 26 inch wheel [17,18], the charging time is between 4 and 6 h [17,18,72], the number of life cycles is between 500 and 1000 [17,18,69], and the overall efficiency is 86% [75]. There are also e-bikes that use SC as a storage element, such as [58,60,74]. SCs have a greater number of life cycles (see Table 5) and the charging time is much lower: below 1 min [74]. In order to comply with the different power regulations for e-bikes throughout the world, mainly in the 250–750 W range, many implementations have considered 250 W as the maximum power delivered to the e-bike, such as references [17,18,72].

Other implementations have considered greater power ratings in order to increase the speed and, thus, the range. In reference [73], for reaching speeds of 25 km/h, the maximum power delivered was

950 W. In reference [76], for a speed of 32 km/h and a slope of 3% the maximum power reached was 1018 W, which is similar to reference [74]. Bigger e-bikes, like the one presented in reference [75], go up to 2000 W. Based on the second approach, we have designed a hybrid e-bike that splits the operation between the SC and the Li-ion battery, as illustrated in Figure 7. For the design of our hybrid e-bike, the ratio between battery and SC capacitances was set in order to compensate for the e-bike's maximum kinetic energy by functional SC capacitance. The performance and economic impact of this implementation is compared to other PEVs and medium BEVs, known also as LEVs, in Table 9 and Figure 8.

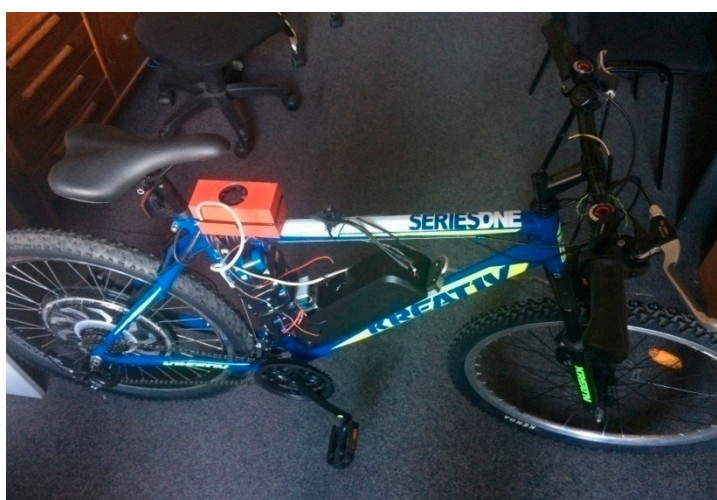

**Figure 7.** Hybrid e-bike.

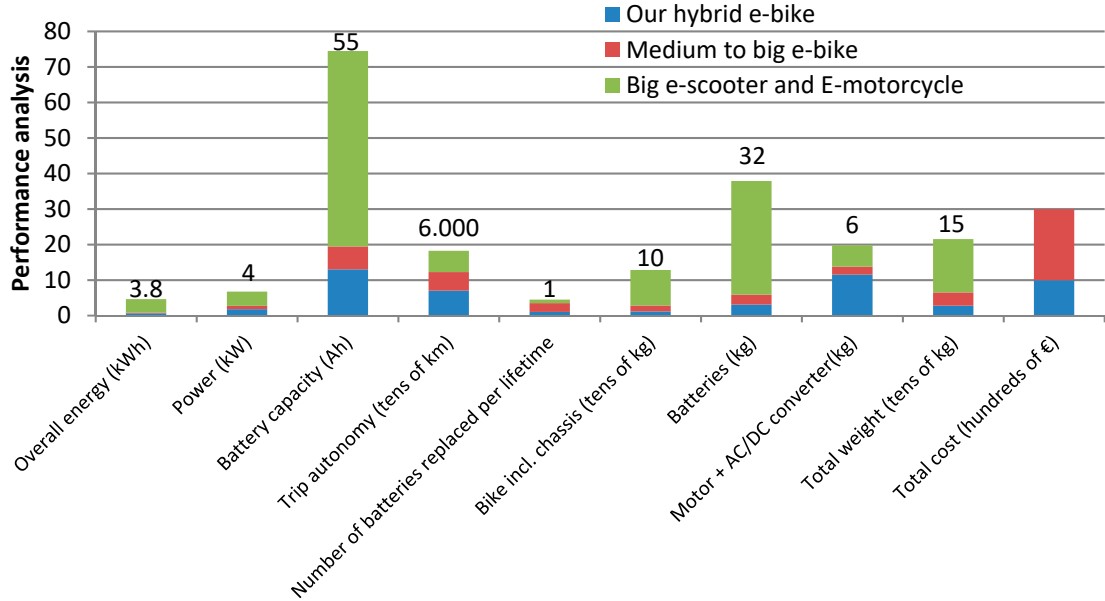

**Figure 8.** Performance analysis and costs of our hybrid e-bike, compared to other PEVs, according to the average values from Tables 8 and 9. The values on top are for big e-scooter and e-motorcycle.

## 6. Discussion

The current state of transportation vehicles has revealed the necessity to develop cleaner electric vehicles especially due to the growing global impact of traffic and air pollution on human health, mainly associated with urban environments, as seen in Table 1. Many EVs have been developed by the industry. Most of these implementations, such as e-cars, try to cover issues like driver safety and

comfort, range, and life span. But the price tag is too high when compared to conventional vehicles. Trip autonomy is very limited in comparison with conventional transportation systems. By employing batteries, this can cause also other problems, like reduced lifespan (10 years, on average). The lack of practical battery recycling solutions leads to this limit in life span. There is also a lot of suspicious advertising related to lifespan. Many consider that their batteries have the best life span, but this is relative due to the sensitivity of these devices and thus leading to a certain level of distrust when taking design decisions. In recycling, the "second life" of batteries is of utter importance, especially for traction batteries. These are used in cheap stationary systems, when the threshold is around 20–30% of the initial nominal capacity.

Other issues include lack of infrastructure and no benefits to fitness. Because 90% to 95% of its time a car is not in operation, in the future, car sharing could be a solution to cut costs and reduce the impact of batteries by splitting it to the number of users. However, until that day, we must look for cheaper and cleaner alternatives for personal transportation vehicles, especially in congested traffic. PEVs represent one such solution, as justified in Table 1. The reduced price of these vehicles also comes with a considerable loss in range, comfort and safety (mainly due to the lack of infrastructure). Yet, many e-bikes have been deployed in China. In reference [81], the environmental impact and safety of e-bikes are compared to that of other transport modes. One can applaud the cost reduction of these lead-acid-based e-bikes and not think about the consequences on environment associated with the lead battery losses [82,83]. We tacklethese issues by means of life cycle assessment (LCA) in Tables 2 and 3. As shown in Figures 4 and 5, the dependence on batteries cuts the benefits of owning a personal electric vehicle, especially for small to medium BEVs, due to their reduced life span (3–4 years). So, even if lead-acid batteries are replaced by other batteries, battery disposal is still a problem for the environment. A much cleaner solution is Li-ion, not only for e-bikes but also for other vehicles such as e-scooters and e-motorcycles, as well as heavier EVs. It is important to mention that LCA tackles the environmental and health impacts and the energy consumption aspects, but it tells nothing about the dynamic performance, especially its sensitivity at wrong charging/discharging cycles or the combination of more negative factors.

In Tables 4–6, we have analyzed the performance metrics and limitations of the main storage solutions for PEVs, which also include SCs and FCs, as alternatives to Li-ion batteries. Energy harvesting (e.g., PV panels) is another option, but this is out of our paper's scope. LCA does provide a certain methodology and thus an almost predictable uncertainty associated with the changes in the driver's behavior and in technology. Yet, in the case of our performance analyses in Tables 4–6, we have observed a large uncertainty. This can be attributed to the great discrepancy between studies, and it can be observed in Tables 4–6 and Tables 8 and 9. For example, in reference [84], the proposed e-bike has a total weight of 111.4 kg. The bicycle and motor weigh 41.4 kg. The payload weighs 43.6 kg, and the batteries (made of 2 Ni-MH 522Wh packs) 26.4 kg. However, most e-bikes are much lighter (usually 20–30 kg). Big e-bikes go up to 60–65 kg. Thus, these e-bikes have a subunit ratio (e-bike mass/rider mass), such as references [85–87]. In these references, a clear distinction should be made between pedelecs and hybrid e-bikes. In reference [88], one can see the limitations of Ni-MH batteries, such as reduced cycle life—only 200–300 cycles. In reference [89], it is stated that battery life of LFP batteries is 10 years. These batteries usually do not exceed 1000 cycles per lifetime. By looking at these examples, we consider that the absence of a standard testing methodology from these articles is one of the main reasons for obtaining such uncertainty. Another reason is the vagueness related to the load demands or charging behavior, random variations of temperature, and overvoltage that can cause significant problems to batteries' reliability and life span. These are still complicated issues, which not many are willing to tackle.

Coming back to dynamic performance, which is another neglected aspect, life expectancy is a result of both the chemical interactions and temperature of operation which affect the batteries. Future research should address issues such as battery aging, and thus underline the sustainability question: Is green also sustainable? This question was also stated in Figure 3. The performance of

battery packs, mostly Li-Ion batteries, are discussed in terms of aging in [55,57,90–93], in terms of temperature impact in [94,95] and also of state of balance/ health [96], in terms of both aging and temperature of operation in [56], in terms of both aging and sizing in [97], and in terms of both aging and energy management/power estimation in [98–100]. Also, the experimental settlement has a strong influence on the outcomes reported, and for that reason we can find out contradictory results in many papers.

In Figure 6 one can observe the main differences between Li-ion batteries, SCs, and FCs in terms of specific energy (Wh/kg) and specific power (W/kg) and get a general idea on how the demands of a PEV in terms of range and acceleration can be fulfilled, which represents mainly the chemist's point of view. If batteries and FCs provide a good autonomy due to high energy, SCs provide fast accelerations due to high power. FCs and batteries cannot supply such accelerations. Various studies have either considered standalone implementations with one of these storage elements (mainly Li-ion batteries for pedelecs) or combinations between two of these three storage elements (as hybrid vehicles). Storage hybridization has already been proven beneficial for larger EVs. Such implementations include tramways [59,61], buses [101,102], and light rail vehicles [103].

An e-bike (based on Li-ion batteries) and a hydrogen bike (based on FC) are compared in Table 7 in terms of LCA. Also, a combination between the two, a hybrid e-bike is compared in the same table with the two standalone storage implementations. Such combinations can be beneficial not only in terms of environmental impact and energy efficiency but also in terms of costs, as several studies have shown. SCs have much better energy efficiency (97–98%) than Li-Ion batteries (86%) and FC (usually 40%), a large number of life cycles (100,000 to 1,000,000), whereas both FC and Li-Ion are modest (500 to 10,000), large power ratings (thousands of W/kg), whereas both FC and Li-Ion go up to hundreds of W/kg, and a fast charging time (seconds) which is incredible when compared to Li-Ion batteries (5 h). Batteries and FCs have a much better specific energy (hundreds to thousands of Wh/kg), whereas SCs do not exceed 10 Wh/kg.

One can see that, when designing a vehicle with HESS, one of the best compromises in overall performance can be found between SCs and batteries, especially in terms of dynamics. Table 8 analyzes the performance metrics of hybrid e-bike implementations compared to other implementations for PEVs—e-bikes, e-scooters, and e-motorcycles. Table 9 proves that our hybrid e-bike presents a good sustainability. We should mention that this was obtained as a result of several measures taken during the design of the system, consisting in replacement of batteries with a hybrid storage solution, adequate sizing of the fast release storage component (SC capacitance) that fully covers the dynamic kinetic energy variation during the e-bike's operation and also appropriate design of the power electronics functions that ensure that the maximum limits are respected by the hybrid storage system.

The sizing process, as shown in the preliminary results obtained for our e-bike, satisfies the reliability and life span requirements foreseen initially. The e-bike's life span depends on exploitation conditions, such as biker's speed profile, daily traveled distance, and thermal behavior of the e-bike's storage system. These interdependences have a major influence on product life cycle (LPC or LLC), reflected by LCA and the performance analyses, as discussed in our paper.

For the design of our hybrid e-bike, the control of power flows is essential for ensuring the system's energy efficiency, by means of power electronics. The majority of applications, such as hybrid e-bike, are related to electric mobility. From the application point of view, the existing standards based on statistical data for speed profile, such as Artemis Cycles, New European Driving Cycle (NDEC), and Worldwide Harmonized Light Vehicle Test Procedure (WLTP), have inherent deviations in reality. In the case of vehicles equipped with ICE these deviations are damped without affecting their reliability. In the case of electric vehicles, the deviations from the "real world" can easily affect the reliability and the lifetime of the EV's energy sources. As mentioned previously, we have observed that the variations in the setting of the experimental test conditions influence the experimental results, sometimes even significantly.

An important remark is related to the necessity to unify the visions of chemists and engineers in the field of electric storage solutions. This should consider not only the power and energy density but also the technology of the devices, packaging, and the control functions of the attached power electronic system. This could lead to a better compromise solution that will increase the sustainability of mobile applications.

At the end of this discussion, we should think also about the changes in customers' habits. By a deep understanding of the necessity to adjust the comfort conditions, an improvement of ecologic footprints should result from using personal electric vehicles instead of electric cars for the transport needs of a single person.

## 7. Conclusions

Due to the shortages in the methodologies used for improving the sustainability of e-mobility (EVs) and lack of solutions for designing hybrid EVs that can offer the right balance (or compromise) between sustainability challenges (economic, societal, and environmental) and performance and technology constraints (dependent on application), the main findings of this paper tackle these issues, consisting in the proposal of an appropriate framework based on LCA and performance analyses for testing the sustainability of EVs, for most sustainability challenges and the sizing of HESS for a hybrid e-bike, based on SC and Li-Ion batteries storage, for optimal personal electric vehicle operation. This can be seen as an e-mobility solution.

The research question we have addressed in the introduction can be easily reformulated—is our hybrid e-bike sustainable? In Table 9, we have analyzed our solution in terms of sustainability, mainly energy consumption, autonomy, costs, and battery disposal. Improvements were observed in all of these categories, plus the life span of such electric vehicle is much better than that of a conventional e-bike from 1–2 years to 2–4 years, as a result of storage hybridization in conjunction with its appropriate sizing.

A compromise was obtained by modifying the ratio between the two storage system component capacitances (battery and SC). This can represent an optimization method for reaching an overall performance of EVs, and implicitly the sustainability goals. We have put in practice this method by developing a hybrid e-bike that is similar in weight to a normal e-bike (<30 kg), but it is able to deliver more power (up to 1800 W, instead of 250–750 W for an e-bike), which is at the same level with a big e-bike that weighs around 60–100 kg. The sustainability goals discussed and delivered by the article were compared to that of other implementations, in order to highlight the strengths of our solution. The methodology we have used was adapted to the application's requirements (hybrid PEVs for e-mobility), paving the way to the deployment of new strategies and procedures necessary to fulfill the sustainability challenges presented herein.

**Author Contributions:** Conceptualization, M.M.-P., and P.N.B.; methodology, M.M.-P.; software, P.N.B.; validation, M.M.-P., and P.N.B.; formal analysis, M.M.-P., and P.N.B.; investigation, M.M.-P., and P.N.B.; resources, P.N.B.; data curation, M.M.-P.; writing—Original draft preparation, M.M.-P., and P.N.B.; writing—Review and editing, M.M.-P., and P.N.B.; visualization, M.M.-P.; supervision, P.N.B.; project administration, P.N.B.; funding acquisition, P.N.B. All authors have read and agreed to the published version of the manuscript.

**Funding:** This research received no external funding.

**Acknowledgments:** We like to thank Transilvania University of Brașov.

**Conflicts of Interest:** The authors declare no conflict of interest.

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
