# Peer review of "Are Personal Electric Vehicles Sustainable? A Hybrid E-Bike Case Study"

_sustainability, doi:10.3390/su12010032_

Round 1

Reviewer 1 Report

The paper is interesting and fits into the journal's scope. Therefore this referee would recommend publication after the implementation of major changes:

Please specify air pollution, kilotons of what over which time period. Please provide energy consumption in addition to MJ/user also in kWh / user and provide also the underlying km-value for each category. This referee doubts the average lifetime for cars. For example in Germany more than 40% of all operating vehicles are older than 10 years. After reaching their lifetime in Germany, they are typically sold to other countries. Source https://www.kba.de/DE/Statistik/Fahrzeuge/Bestand/Fahrzeugalter/fahrzeugalter_node.html Also the lifetime mileage for cars is way to low, e.g. General Motors in the US provides a warranty of 100,000 miles (160,000 km) for EVs. An EV truck of course has limitations in speed and cargo capability, please add. Range value is way too high for EVs: Today typically 200 and 300 km WLTP range per charge, in future 500 km per charge. Charging time of EVs is by no means neglible, its 30 min up to several hours for a large EV battery, even when using ultra-fast charging tech.

Line 191 ... concerning alternatives to BEVs, please add also range-extender EVs. A potential source might be.

R. Matthé, U. Eberle, The Voltec System: Energy Storage and Electric Propulsion, In book: LITHIUM-ION BATTERIES: ADVANCES AND APPLICATIONS, Edition: 1, Chapter: The Voltec System: Energy Storage and Electric Propulsion, Publisher: Elsevier, Editors: Gianfranco Pistoia, pp.151-176, DOI: 10.1016/B978-0-444-59513-3.00008-X

Author Response

Thank you very much for your suggestions and remarks. It was very helpful for us in order to improve our paper.

We have updated our paper according to your remarks. Our answers are:

1) Please specify air pollution, kilotons of what over which time period.

We have mentioned the Table 1 legend that this quantity is related to the main pollutants: CO, NOx, VOC, and PM10. The time period is one year (it is mentioned in the table: kilotons/year)

2) Please provide energy consumption in addition to MJ/user also in kWh / user and provide also the underlying km-value for each category.

We have adapted our Tables 1, 2 and 9 in accordance to your observations, and we have kept both units (MJ and Wh, and Wh/km)

3) This referee doubts the average lifetime for cars. For example in Germany more than 40% of all operating vehicles are older than 10 years. After reaching their lifetime in Germany, they are typically sold to other countries. Source https://www.kba.de/DE/Statistik/Fahrzeuge/Bestand/Fahrzeugalter/fahrzeugalter_node.html

We have modified in Table 1 the life span for cars from 10 years to 10-20 years.

4) Also the lifetime mileage for cars is way to low, e.g. General Motors in the US provides a warranty of 100,000 miles (160,000 km) for EVs. An EV truck of course has limitations in speed and cargo capability, please add.

In Table 1 we have changed the value to 160 000 km for cars. For trucks the life span is around 500 000 – 1 000 000 km. In case of EV trucks the life expectancy depends on infrastructure. We considered only EVs powered by batteries in this case.

5) Range value is way too high for EVs: Today typically 200 and 300 km WLTP range per charge, in future 500 km per charge.

We have changed the range value for EVs to a few hundreds of km, which is accordance with your remark. We have changed the maximum value to 500 km for EVs.

We have commented the following in Discussions (chapter 6):

- regarding second life of batteries:

“Many consider their batteries with best life span, but this is relative due to the sensitivity of these devices, and thus leading to a certain level of distrust when taking design decisions. In recycling, the “second life” of batteries is of utter importance, especially for traction batteries.”

- regarding lifetime and driving cycles

“The majority of applications are related to electric mobility. From the application point of view, the existing standards based on statistical data for speed profile, such as: Artemis cycles, New European Driving Cycle (NDEC) and Worldwide Harmonized Light Vehicle Test Procedure (WLTP), have inherent deviations in reality. In the case of vehicles equipped with ICE these deviations are damped without affecting their reliability. In the case of electric vehicles, the deviations from the "real world" can easily affect the reliability and the lifetime of the EV’s energy sources.”

6) Charging time of EVs is by no means negligible, it is 30 min up to several hours for a large EV battery, even when using ultra-fast charging tech.

In Table 1 we have changed the value of recharging time for EVs from a few hours to tens of minutes to a few hours. We have mentioned below Table 1 that this time is “ ** Dependent on charger and battery type”

7) Line 191 ... concerning alternatives to BEVs, please add also range-extender EVs. A potential source might be.

We have considered your remark and added in text the following:

“The demand for energy storage on board has led to the increase in the standard voltage of the electrical energy sources energy per vehicle from 12 or 24V to 48V, respectively above hundreds of volts for pure electric vehicles. Also plug-in electric cars and vehicles with increased range when operating in electric regime have been proposed. The latter, which are known as or "extended ranger" systems, can increase the range of the vehicle, but with a sacrifice in nominal performance [13].”

It can be found in the References section:

“13.  Matthé, R., & Eberle, U. The Voltec System-Energy Storage and Electric Propulsion. In Lithium-Ion Batteries: Advances and Applications, 2014. https://doi.org/10.1016/B978-0-444-59513-3.00008-X”

Reviewer 2 Report

This work presents an extensive literature review on electric vehicles and particularly on e-bikes features, such as, energy consumption, environmental impact, and storage solutions. The authors provided a well-structure paper that is easy to follow. However, I have only a couple of observations:

1. Table 8. Can the authors make a chart out this data? This will provide to the reader an easy way for digesting such information.

2. Provide a y-axis legend in Figure 2, 3, and 6.

3. Could you elaborate on the impact of this transport policy recommendation? “….for developing the e-bike of 2050, less lithium must be used per unit …. ”

Author Response

Thank you very much for your review.

We updated our paper according to your remarks. Our answers are:

1. Table 9 integrates many of the data from Table 8. Therefore Figure 8 in the updated paper contains both data from these two tables. We adapted the name of the Figure 8 in accordance.

2. We have provided the Y-axis legend for Figures 4,5 and 8 (which are the new versions of Figures 2,3 and 6, which were the initial figures in the original paper)

3. We have discussed the implication of this policy in our paper in the lines below:

“As recommended in [25], for a large scale deployment of EVs, less lithium must be used per unit of battery storage, or a suitable type of energy storage system that does not use lithium must be developed. This is justified by the cumulative demands of EV/PHEVs that could exhaust the whole lithium reserve by 2050, even with extensive recycling”.

Also in Discussions (chapter 6, in the new version of the paper) we have emphasized on the importance of reducing the Li-Ion battery dependency by hybridization of the storage system with supercapacitors and the adequate sizing of the components:

“Table 9 proves that our hybrid e-bike presents a good sustainability. We should mention that this was obtained as a result of several measures taken during the design of the system, consisting in: replacement of batteries with a hybrid storage solution, adequate sizing of the fast release storage component (SC capacitance) that fully covers the dynamic kinetic energy variation during e-bike operation, and also appropriate design of the power electronics functions that assure that the maximum limits are respected by the hybrid storage system“.

Reviewer 3 Report

Apart from language inefficiency, manuscript suffers from poor structuring in all sections. For instance, the abstract should have the same structure as the entire manuscript. In the introduction authors have failed to set a strong background of the known knowledge. The authors only described the problem but did not provide the state of preparation on the world background.

Moreover, the writing is not always easy to follow for the reader, especially since there is sometimes no logical transition from one paragraph to another.

Conclusions:

Although mostly pertinent to the study, I have some observations:

Firstly, conclusion is excessively long (needs to be more synthetic and self-explicative)

Secondly, in some points the stated exceeds the scope of the data and, given the lack of support in some findings along the discussion, I believe it should be revised and re-elaborated.

Limitations:

The study has several both limitations and strengths, that (I believe) are simply not stated, except for some parts of the document. Authors must put an additional effort on identifying, describing and discussing these issues.

Also:

Some "clean up" of the language is needed. It can be useful to seek some assistance with finalizing the language in such a paper prior to submission or re-submission. The paper could be rearranged or use a tighter focus. Perhaps the authors might re-formulate their argument and focus on adding a new perspective to this area or further developing this area? A short, introductory paragraph summarizing the intent and scope of the study would provide a useful context for the rest of the paper. In addition, a closing paragraph or two with summative insights, or overarching principles garnered from this historical review would make the manuscript more complete. As written, the final paragraphs seem an abrupt end to the work and the paper lacks a sense of closure.

Author Response

Your suggestions and remarks were very useful for us in order to reorganize our paper and reformulate the numerous ideas and issues we have tackled in the original version. In this sense, we have updated many parts of our paper. Our main changes in the new version of the paper are synthesized below:

- The Abstract was reformulated to better suit the scope of the paper. It now reflects the paper’s structure. The main findings of our paper are now correlated with the final chapter (Conclusions).

- The Introduction now provides a corresponding background in the context of sustainable transportation: we added two new Figures (Fig. 1 and 2) correlated with Figure 3 (the old Figure 1), regarding the methodology we have used in the paper for addressing the sustainability challenges.

- Chapters 2-5 were adapted accordingly to the new changes in Abstract and Introduction, and thus underline the main findings of our paper.

- Discussions (chapter 6) reiterate the main findings of our paper, and offer the necessary details for understanding the issues tackled in the paper and also the issues related to future research on sustainable transportation

- Conclusions The final chapter was reduced to three main paragraphs which state the main findings of our paper: - new methodology for testing sustainable transport systems (EVs) and - a solution for developing PEVs, based on hybrid storage (our hybrid e-bike)

Now we can offer detailed answers related to these changes, by taking into account your remarks:

1) Apart from language inefficiency, manuscript suffers from poor structuring in all sections. For instance, the abstract should have the same structure as the entire manuscript.

I agree with your remark, that language inefficiency is problematic in case of our paper. Unfortunately, English is not our native language. We believe that we have improved this aspect in the new version. Still, we have presented most of the issues in a coherent fashion, according to the structure presented in the new abstract, which now has the same structure as the entire manuscript.

2) In the introduction authors have failed to set a strong background of the known knowledge.

We have improved the Introduction section by organizing it in a more logical and hierarchical way, and hopefully, easier to follow by means of new Figures 1-2, which lead to Figure 3. We included more information regarding the context of our research efforts, starting from the sustainable transportation context, describing the relations of the transport systems with the sustainability challenges, and the current state of transportation systems from a historical point of view. Also, the text was adapted in order to reflect the methodology proposed in the paper, in case Figures 1-3 do not offer the complete image. We believe that the knowledge background can be correlated with Tables 1-9 and Figures 1-8 and with the references, which should offer sufficient information regarding the current issues of sustainable transportation, mainly in the case of PEVs.

3) The authors only described the problem but did not provide the state of preparation on the world background.

In the new Introduction chapter, we have offered some clarifications regarding the three main paragraphs of the Abstract:

Abstract (1st paragraph): “As the title suggests, the sustainability of personal electric vehicles is under question. In terms of life span, range, comfort, and safety, electric vehicles, such as e-cars and e-buses, are much better than personal electric vehicles, such as e-bikes. But, electric vehicles present greater costs and increased energy consumption. Also, the impact on environment, health and fitness is more negative than that of personal electric vehicles”.

The context is presented in Introduction, mainly by means of Figure 1 and the paragraphs describing the sustainability challenges of current transport systems (EVs). The history and classification of EVs is also presented in Introduction.

Abstract (2nd paragraph): “Since transportation vehicles can benefit from hybrid electric storage solutions, we address the following question: Is it possible to reach a compromise between sustainability and technology constraints by implementing a low cost hybrid personal electric vehicle with improved life span and range that is also green?”

The possibility to reach a compromise between sustainability challenges and technology (/performance) constraints can be materialized by means of hybridization. The need to find such compromise is a result of the information presented in Table 1.

Abstract (3rd paragraph): “Our methodology consists of life cycle assessment and performance analyses, asserting the facets of the sustainability challenges (economy, society, and environment) and limitations of the electric storage solutions (dependent on technology and application) presented herein”.

Figures 2-3 from the new Introduction present the methodology used in our paper for addressing the guidelines that aim to find the right balance (or compromise) between the design of our transport solution (in our case, a hybrid e-bike), mainly HESS sizing, and sustainability challenges: LCA and performance analyses. In the next chapters (2-5), we demonstrate that a HESS with SCs and Li-Ion batteries provides the best compromise. In Discussions and Conclusions, we look back at the methodology and our solution. The shortage of relevant test methodology and sustainable solutions for PEVs was the main reason for conducting our study. We discuss the current research trends in sustainable transportation, and think also about the future trends in Discussions. A deep understanding of the issues affecting sustainable transportation can lead to applicable solutions in the future. We believe that our hybrid e-bike is a starting point.

4) Moreover, the writing is not always easy to follow for the reader, especially since there is sometimes no logical transition from one paragraph to another.

Again, I agree with your remark. We have tried to do our best, but the issues we have tackled in our paper lead us in many directions, that sometimes did not fit the scope of our paper, nor its logic. I hope that the new version will lead our paper in complementary directions, and thus be easier for the reader to follow.

5) Firstly, conclusion is excessively long (needs to be more synthetic and self-explicative)

The final chapter was filtered and now contains only the relevant findings of our paper

6) Secondly, in some points the stated exceeds the scope of the data and, given the lack of support in some findings along the discussion, I believe it should be revised and re-elaborated.

Conclusions were reformulated accordingly to your remark.

7) The study has several both limitations and strengths, that (I believe) are simply not stated, except for some parts of the document. Authors must put an additional effort on identifying, describing and discussing these issues.

I appreciate this remark. We have tried to solve the limitations regarding the paper’s objectives, findings and structure and highlight its strengths, which are related mainly to its main findings. They were described thoroughly in the text, especially in the tables and figures.

8) Also: Some "clean up" of the language is needed. It can be useful to seek some assistance with finalizing the language in such a paper prior to submission or re-submission.

By a better organization of our paper, we have aimed to clarify our statements and filter out the unnecessary sentences.

9) The paper could be rearranged or use a tighter focus. Perhaps the authors might re-formulate their argument and focus on adding a new perspective to this area or further developing this area?

We sought to provide a tighter focus to our paper. In the new version of our paper, the abstract was reformulated in order to comply with the paper’s structure. The objectives and findings of our paper are debated in Discussions also in terms of future development. Future research based on a clear perspective is of utter importance for developing the vehicles of tomorrow. Hybridization brings a new and interesting perspective to PEVs.

10) A short, introductory paragraph summarizing the intent and scope of the study would provide a useful context for the rest of the paper.

In Introduction, we have added a few paragraphs in which we state the problem of sustainability in the context of future transportation systems (EVs to PEVs). In this context we discuss EVs and how their technology and performance limitations can be balanced in order to comply with the sustainability challenges. It can be seen in the Abstract and Introduction that our research intentions are correlated with the current state of transport systems and the sustainability challenges which they try to address.

11) In addition, a closing paragraph or two with summative insights, or overarching principles garnered from this historical review would make the manuscript more complete. As written, the final paragraphs seem an abrupt end to the work and the paper lacks a sense of closure.

A historical review of EVs is presented in Introduction. The Conclusions chapter synthesizes the main findings in the first paragraph. The last two paragraphs reformulate this study’s main findings, and can be seen as an extension of the abstract and Introduction.

The last two paragraphs of the abstract, as stated below, can now be compared with the last two paragraphs of the Conclusion:

Abstract (4th and 5th paragraphs): The hybrid electric storage system of the proposed hybrid e-bike is made of batteries, supercapacitors and appropriate power electronics, allowing the adequate control of power flows between the system’s components and application’s actuators. Our hybrid e-bike costs less than a normal e-bike (half or less), does not depend on battery operation for short periods of time (a few seconds), has better autonomy than most personal electric vehicles (more than 60 km), has a greater life span (a few years more than a normal e-bike), has better energy efficiency (more than 90%) and is much cleaner due to the reduced number of batteries replaced per life time (1 instead of 2 or 3).

Round 2

Reviewer 1 Report

Please check language carefully. This referee has the impression language quality has decreased due to extensive editing. At several points, blank spaces between words, especially in tables. Why do you spend so much effort with the life cycle of lead acid batteries, does not play a role for a traction battery any more. On the other hand there is "no harm" due to that section in case you want to keep it.  This referee doubts that Wouk's Skylark is first hybrid vehicle, although nevertheless interesting. Please mention that it was a private conversion from a production vehicle. GM had earlier own plug-in hybrid for example, see GM XP-883 in 1969. But even way earlier, in the era of the atomotive founding fathers, Porsche constructed a hybrid vehicle, see the Lohner Porsche.

Author Response

Thank you for your feedback. Our answers are the following:

1) Please check language carefully. This referee has the impression language quality has decreased due to extensive editing. At several points, blank spaces between words, especially in tables.

We have corrected and polished many paragraphs, by performing a complete round of proofreading to our paper.

2) Why do you spend so much effort with the life cycle of lead acid batteries, does not play a role for a traction battery any more. On the other hand there is "no harm" due to that section in case you want to keep it.  

Indeed, the technologies based on Lead-acid batteries can be considered obsolete at this moment, but a lot of implementations for transport still exist, mainly in China and in less developed countries. Alloying elements such as Titanium can improve significantly the lifetime and the main properties of Lead-acid batteries. The problem is still open because of the limited resources of Lithium, which is considered the better alternative in terms of energy density and eco footprint. We have mentioned this issue in our paper:

“As recommended in [25], for a large scale deployment of EVs, less lithium must be used per unit of battery storage, or a suitable type of energy storage system that does not use lithium must be developed. This is justified by the cumulative demands of EV/PHEVs that could exhaust the whole lithium reserve by 2050, even with extensive recycling.”

Also, alkaline and manganese oxide elements can be used as basic or alloying materials for batteries, but studies are in development. Materials based on structured matter can provide a new level of energy density for the future battery’s implementations (see graphene or carbon nanotube based materials).

3) This referee doubts that Wouk's Skylark is first hybrid vehicle, although nevertheless interesting. Please mention that it was a private conversion from a production vehicle. GM had earlier own plug-in hybrid for example, see GM XP-883 in 1969. But even way earlier, in the era of the atomotive founding fathers, Porsche constructed a hybrid vehicle, see the Lohner Porsche.

Thank you very much for your remark. We have reconsidered the history of EVs and PHEVs, and adapted it in the new Introduction:

“In 1901 Ferdinand Porsche invented the first hybrid electric car, Lohner-Porsche Mixte. Since 1969, General Motors is preoccupied on developing hybrid driving cars (see GM XP-883 in 1969), and in 1973 they proposed a hybrid car prototype due to pollutions issues, based on the previous model for Buick Skylark. This prototype was designed by Victor Wouk”

Reviewer 3 Report

Authors have made all required changes. However, please do another round of proof-reading.

Author Response

Thank you for your feedback.

We have corrected many paragraphs, by performing a complete round of proofreading to our paper.